# Professionalization of Online Gaming? Theoretical and Empirical Analysis for a Monopoly-Holding Platform

**Vitor Miguel Ribeiro** [1,2,*] and **Lei Bao** [3]

1   Economic Sciences and Tourism Research Center of Consuelo Vieira da Costa Foundation, Higher In-stitute of Administration and Management, 4100-442 Porto, Portugal
2   Department of Economics, Faculty of Economics, University of Porto, 4200-464 Porto, Portugal
3   School of Economics and Management, Huaiyin Normal University, Huaian 223001, China; 8202011046@hytc.edu.cn
*   Correspondence: vsribeiro@fep.up.pt; Tel.: +351-225-571-100

**Abstract:** We analyze the private equilibrium of a two-sided market representing the online gaming industry under a principal-agent model. A monopoly-holding platform hires a manager to attract new members from both sides of the market while considering uncertainty on the adhesion of viewers and online gamers. First, we mathematically demonstrate that increasing cross-group network externalities can decrease the platform's profit, which contradicts a canonical result from the field of two-sided markets. Moreover, knowing that the intermediary's goal is aligned with the private interest of online gamers, machine learning models empirically show that the main theoretical outcome is observed in reality due to the presence of heterogeneous indirect network effects in online gaming activities. Second, we conclude that social welfare can be either harmed or improved for increasing cross-group network externalities, which means that the professionalization of online gaming may or may not be legitimized depending on the value taken by exogenous parameters related to the platform's uncertainty on the number of agents that get on board, risk aversion of viewers, and royalty rate applied to online gamers. Finally, a discussion based on 2020 facts is provided and several policy recommendations are formulated to ensure the persistence of best regulatory practices.

**Keywords:** two-sided markets; monopoly; online gaming; salesforce compensation; machine learning

## 1. Introduction

In 2019, the World Health Organization (WHO) elaborated a report confirming that over 150 million people play online video games on a regular basis in the United States of America (USA). Although some of these individuals are teenagers, Limelight [1] indicates that the average American online gamer is 35 years old and 72% are older than 18 years. A study focused on the People's Republic of China (PRC), Gan [2] shows that females are more active in social media despite males' domination in the online gaming industry. In general, the specialized literature analyzes the impact of online gaming on multiple domains (e.g., psychological, neurological, social) of the individual and two main types of users are identified: viewers and online gamers. Some studies indicate that online entertainment can act in strategic complementarity with traditional jobs based on the argument that individuals benefit from the gaming experience while retaining the capacity to move away completely satisfied when deemed necessary [3]. Their consumption habits are sustained with friends or alone, but the prioritization of real-life targets (e.g., school grades) is usually beyond doubt. At the individual level, this type of user maintains strong family relationships and attends to school and/or office on time. Other leisure activities belong to their daily routines, which suggests that online gaming does not correspond to a make-or-break activity [4]. Yet, other studies show that individuals can fall in the domain of strategic substitutability [5]. However, they do not exhibit a consensual

agreement on short- and long-term effects. A strand of research suggests that online gaming is a vehicle for the promotion of violence and addiction due to the presence of video games characterized by first-person shooting, which can promote a negative impact on the development of synaptic connections in the human brain [6]. In particular, Bloomberg [7] describes PRC as the "2017 games industry capital of the world", whereas the WHO [8,9] emphasizes that the largest E-sports games market can be found there. To retaliate against negative consequences resulting from the engagement of individuals in the online gaming experience, PRC recently imposed several regulatory obligations [10]. Differently, another strand of research presents evidence of users that substitute traditional jobs with online gaming activities but, nevertheless, are diagnosed as being completely healthy. Indeed, recent applications such as Serious Games and Virtual Reality (SGVR) are alternative tools to support rehabilitation therapies, which suggests that the professionalization of online gaming can be seen as a credible threat in the sense of Kreps and Wilson [11].

Considering that a reasonable doubt exists on the persistence of either substitutability or complementarity effects between the maintenance of traditional jobs and the engagement in online gaming activities, a motivating research question is to identify market conditions where the professionalization of online gaming activities can promote welfare-enhancing effects. To analyze such a relevant topic, contrary to the abovementioned contributions that are exclusively centered on the individual, this study takes a societal level approach by relying on the theory of two-sided markets.

### 1.1. Relevant Literature Focused on a Societal Approach: Two-Sided Markets and Salesforce Compensation

Classical studies in the field of two-sided markets start by analyzing the impact of cross-group network externalities on prices and profit [12]. In the case of a monopoly market structure, a first seminal result confirms that cross-group network externalities increase the prices and profit of the intermediary [13]. A second key outcome shows that a monopoly-holding platform has the incentive to distort the pricing structure by applying a 'divide-and-conquer' strategy, which consists of charging a price below (above) marginal cost to the side with superior (reduced) ability to attract agents from the opposite side of the market, respectively. This business initiative allows the attraction of agents from the side whose demand is more price-sensitive (i.e., the 'strongest' side of the market) since these have the capacity to attract agents from the opposite side of the market (i.e., the 'weakest' side of the market) due to their stronger indirect network effect.

The video game industry can be viewed as a two-sided market when considering that one group corresponds to users, while the opposite side of the market is composed of software developers [14]. If restricting the focus on online gaming, one can alternatively think of a two-sided market composed of a group of viewers, while the other side consists of online gamers. Both groups are independent and can only interact through a certain platform (e.g., Twitch), which defines the basic elements of a two-sided market. By taking this perspective as a given, online gaming corresponds to a niche industry leveraged by the formation of network effects, which can have either a direct or indirect nature. A sufficient condition for the prevalence of tipping (i.e., market concentration of agents from both sides of the market into a single platform) is the presence of sufficiently strong indirect network effects. Moreover, a monopoly-holding platform can be sustained over time through the celebration of managerial contracts aimed at promoting membership concentration [15].

Unsurprisingly, the video game industry has been analyzed in recent years by the literature of salesforce compensation, which has dedicated efforts to examining the strategic interaction between platforms, viewers, and online gamers by considering that their relationship is captured by two-sided markets theory [16]. This is particularly remarkable given that online purchases and advances in information technologies have rapidly changed the business landscape [17]. In particular, the biennium 2019–2020 is characterized by shifts in online gaming behavior and new gaming options provided by platforms such as Twitch and Google Stadia, thereby suggesting that a new category of online gaming, designated by 'console-less gaming services', is currently on the rise. Moreover, the binge-

gaming trend has been leveraged by mobile applications (i.e., mobile apps), which have become increasingly popular in developed countries and in most developing markets due to the extraordinary growth of the smartphone market. Consequently, a new economic landscape entitled the App economy has emerged [18], which has fostered a trend favoring the professionalization of online gaming activities.

This strand of literature indicates that online platforms frequently design a principal-agent contract to elaborate a compensation scheme for a salesperson (i.e., manager), whose duty is to internalize not only the agency problem but also the issue of risk-sharing. Under this circumstance, active selling is a key instrument used by managers to develop an appropriate business plan to conquer additional market share, which suggests that combining two-sided markets and salesforce compensation literatures is justified by the fact that the optimal compensation plan offered by platforms to managers and the effort spent by managers to get additional 'members on board' do not neglect that markets where online platforms currently operate are composed of agents that exhibit heterogeneous characteristics, thereby constituting elements that define different sides of a complex intermediation system that accommodates distinct indirect network externalities.

As such, a key concern from the platform's perspective is not necessarily the negative impact of online gaming on the health of gamers, but the uncertainty on the number of viewers and online gamers that join the platform. Traditionally emerging from seminal moral hazard studies [19–21], this type of market environment reflects that a monopoly-holding platform has to deal with demand uncertainty on both sides of the market after the incentive contract being signed with the manager. In the context of two-sided markets, moral hazard has been covered in multiple situations. When dealing with the need to keep a platform's reputation beyond doubt, Roger and Vasconcelos [22] demonstrate that prices can be used as simple and effective tools to mitigate the opportunistic behavior of sellers when combined with the threat of exclusion. In particular, platforms can adopt registration fees to keep low-quality sellers (i.e., those more prone to behave opportunistically) out of the market. Querbes [23] considers a platform that governs the diffusion of information between buyers (i.e., purchases and reviews) and sellers using an agent-based simulation model. From an intermediation perspective, a market failure is more likely to hold when participants are only extrinsically motivated, and when control is either too constraining or completely null. From the standpoint of market participants, categories and transactions that should be banned are very specific: the low-quality demand is unable to access the marketplace due to overpricing of the low-quality supply (i.e., similar conclusion to the one observed in the market for lemons) and the high-quality supply is undervalued (i.e., similar conclusion to the distortion at the top principle). Recent studies of the business and marketing literature also reinforce the relevance of network effects on the development of new online gaming niches [24–27]. Contemporaneous debates on Internet gaming disorder can be found in Ref. [28–31]. An exhaustive compilation of studies focused on analyzing the online gaming industry is exposed in Ref. [32,33].

### 1.2. Research Objectives and Main Results

This study consists of developing a principal-agent model in the context of a two-sided market by relying on a game theory approach, where a monopoly-holding platform hires a manager to get agents from both sides on board while assuming uncertainty on their adhesion to the platform. This study aims at filling a research gap related to the share and debate of new ideas in a rapidly evolving online two-sided market whose business practices, social, cultural, and legal concerns, personal privacy and security, and social welfare effects are still unknown. Societal issues related to the professionalization of online gaming are among the most relevant concerns of a new digital age characterized by disruptive employment opportunities compared to the period before the rise of the Internet of Things (IoT), which may or may not act in the benefit of society. Hence, we propose to satisfy four main research objectives.

First, we aim to characterize the subgame perfect Nash equilibrium (SPNE) of the static game where a platform defines the price to be charged to viewers and the compensation plan to be offered to the manager before the manager applying effort to ensure that additional members join the monopoly-holding platform and participate in the market.

Second, knowing that the platform's commercial goal is perfectly aligned with the private interest of online gamers, we aim to empirically validate the theoretical outcome sustained after satisfying the first research objective. This study demonstrates that the surplus enjoyed by online gamers can decrease when the indirect network effect promoted by this group of agents on the opposite side of the market becomes more intense. Knowing that a similar conclusion is applied to the profit enjoyed by the monopoly-holding platform, this study finds opposite conclusions to those observed in Armstrong [13], according to which cross-group network externalities unambiguously have a positive effect on the intermediary's profit. Furthermore, the presence of a negative indirect network effect is proven both from a theoretical and empirical point of view.

Third, we assess the impact of both cross-group network externalities on social welfare and contextualize respective findings in light of recent events affecting online platforms such as Twitch, YouTube, Twitter, and Facebook to understand whether the growth of indirect network effects and the trend in favor of the professionalization of online gaming are socially desirable actions. By doing so, we provide useful policy recommendations aimed at ensuring the persistence of best regulatory practices. This study shows that the professionalization of online gaming, which is captured by increasing cross-group network externalities in both sides of the market, can either harm or improve social welfare. In particular, we confirm that increasing cross-group network externalities in both sides of the market have undesirable effects on the level of social welfare when the following cumulative conditions hold: when both cross-group network externalities are sufficiently strong, when the demand uncertainty in both sides of the market is negligible, when a well-established audience of viewers is not excessively averse to the online gaming experience, and when the royalty rate imposed by the monopoly-holding platform to online gamers is not too high. Under this market circumstance, the professionalization of online gaming implies a social welfare loss and, analogously, it justifies actions recently observed in reality (e.g., censorship of content at YouTube, Facebook and Twitter).

Finally, we evaluate the impact of the asymmetric gap between cross-group network externalities on the equilibrium value of the managerial compensation provided by the platform and effort spent by the manager. To satisfy this final purpose, the mathematical framework presented in this study extends the existing literature of two-sided markets and salesforce compensation. In particular, three improvements are provided relatively to Ribeiro [15], which is the study closest to ours: (I) imposition of the endogenous price charged to viewers with uncertainty rather than perfect information; (II) development of empirical analysis to validate the theoretical result applied to online gamers, and; (III) contextualization of findings related to the impact of cross-group network externalities on social welfare. Research questions associated with the study's content can be summarized as follows:

**RQ1.** *Which impact does the presence of uncertainty on the number of agents that join the monopoly-holding platform in both sides of the market have on equilibrium outcomes?*

**RQ2.** *How does the asymmetric gap between indirect network effects influence the compensation plan developed by the principal and the managerial effort spent by the agent?*

**RQ3.** *Can social welfare enhance with increasing membership in the online gaming industry? If so, under which market conditions?*

The rest of the study is organized as follows. First, we present a formal theoretical model. Second, we perform a mathematical analysis. Third, we expose the main results. Fourth, we examine the surplus enjoyed by each side of the market and provide empirical

validation of the main theoretical outcome previously found. Thereafter, we infer and discuss the impact of cross-group network externalities on social welfare. Finally, we summarize the main conclusions. Supplementary Material supports the main text for the sake of brevity.

**2. Method**

Consider the representative model of a two-sided market composed of viewers that interconnect with online gamers through a platform (i.e., principal) that delegates the operational activity (i.e., membership attraction or sales enforcement) to a manager (i.e., agent). The model is characterized by demand uncertainty faced by the monopoly-holding platform in both sides of the market given that one assumes that the number of viewers and online gamers that join the platform is unknown after the realization of the managerial contract. Hence, the compensation plan offered by the platform and managerial effort has influential power on the ex-post level of market participation.

*2.1. Demand Side*

The indirect utility of a representative viewer that becomes a member of the platform to consume content is given by

$$U = v - p + \theta n_d + e \tag{1}$$

where $v$ is the standalone value without interaction with the opposite side, $p$ corresponds to the access price charged by the platform, $n_d$ is the amount of content, $\theta > 0$ is the intensity of the cross-group network externality (i.e., an indirect network effect) exerted by an additional unit of content created by online gamers on the representative viewer, and $e$ represents the managerial effort [34]. Without loss of generality (W.l.o.g.), assume that $v$ follows a uniform distribution in [0, 1]. A representative viewer indifferent between joining the platform and staying out of the market is formally given by

$$U = 0 \Leftrightarrow v^* = p - \theta n_d - e \tag{2}$$

Assuming that a demand shock follows $\varepsilon_u \sim N(0, \sigma_u^2)$, the total number of viewers is given by

$$n_u = \int_{v^*}^1 1 \, dx \Leftrightarrow n_u = [x]_{v^*}^1 \Leftrightarrow n_u = 1 - p + \theta n_d + e + \varepsilon_u \tag{3}$$

Equation (3) shows that the adhesion of viewers to the platform depends on the price, cross-group network externality of online gamers on viewers, managerial effort, and the error term $\varepsilon_u$.

*2.2. Supply Side*

We assume that one additional unit of content consumed by viewers provides a benefit $g > 0$ to the representative online gamer. Moreover, we consider that the representative online gamer must pay a royalty rate $\gamma$ to the platform to ensure that it remains active in the market, $0 < \gamma < 1$. Considering full market coverage such that viewers absorb all the available content in the platform, the revenue of a representative online gamer is given by

$$R_d = g n_u (1 - \gamma) \tag{4}$$

The representative online gamer has a fixed cost $f$, which is assumed to be uniformly distributed in the unit interval, thereby implying that it obtains a surplus given by

$$\pi_d = R_d - f \Leftrightarrow g n_u (1 - \gamma) - f \tag{5}$$

Let $f^*$ be the marginal online gamer who is indifferent between playing and not playing games on the platform. Formally

$$\pi_d = 0 \Leftrightarrow f^* = gn_u(1 - \gamma) \tag{6}$$

Assuming that a supply shock follows $\varepsilon_d \sim N(0, \sigma_d^2)$, the total number of online gamers is given by

$$n_d = \int_0^{f^*} 1 \, dx \Leftrightarrow n_d = [x]_0^{f^*} \Leftrightarrow n_d = gn_u(1 - \gamma) + \varepsilon_d \tag{7}$$

Equation (7) reflects that the adhesion of online gamers to the platform depends on the royalty rate, cross-group network externality of viewers on online gamers, and the error term $\varepsilon_d$. Both $\varepsilon_u$ and $\varepsilon_d$ are assumed to be independent and identically distributed.

### 2.3. Managerial Compensation and Effort

Following Ref. [35], a linear compensation plan $S(n_u)$ is determined by the platform to motivate the manager on applying effort to attract new members

$$S(n_u) = \alpha_0 + \alpha_1 \, p \, n_u \tag{8}$$

Equation (8) reveals that the compensation plan consists of two components: a fixed salary $\alpha_0$ and a variable bonus $\alpha_1 \, p \, n_u$ that depends on the commission rate $\alpha_1$ and revenue $p \, n_u$ extracted from viewers. As such, the incentive design is said to be 'one-sided revenue-based' given that the managerial compensation is strictly based on the mobilization of viewers (i.e., agents that only consume content or, similarly, without any kind of responsibility on content creation). Following Ref. [20], the manager is assumed to be risk-averse such that the respective utility is given by

$$\Phi = 1 - e^{-\rho[S(n_u) - C(e)]} \tag{9}$$

where $\rho$ corresponds to a measure of risk aversion faced by the representative viewer. The manager obtains a positive utility from the compensation plan $S(n_u)$, but holds a negative utility from the additional effort spent on membership attraction. Let $C(e)$ denote the effort cost function, which satisfies $C'(e) > 0$ and $C''(e) > 0$. Similar to Ribeiro [15], $C(e) = e^2/2$ is assumed. Given the presence of bilateral demand uncertainty, the certainty equivalent of the manager's utility function is given by

$$U_{CE} = E[S(n_u)] - (\rho/2)\text{Var}[S(n_u)] - C(e) \tag{10}$$

The effort level is determined by maximizing Equation (1), which corresponds to the incentive compatibility (IC) constraint. The individual rationality (IR) constraint also must be satisfied, according to which the manager obtains a non-negative utility. W.l.o.g., the outside option is normalized to zero.

### 2.4. Profit, Surpluses, Social Welfare, and Timing Structure

The platform defines the managerial contract bearing in mind the maximization of expected profit

$$E[\Pi] = p \, E[n_u] + \gamma \, gE[n_d] - E[S(n_u)] \tag{11}$$

subject to the fulfillment of IC and IR constraints. The surplus of the representative viewer is given by

$$CS_u = \int_v^1 x \, dx \Leftrightarrow CS_u = \left[x^2/2\right]_v^1 \Leftrightarrow CS_u = (1 - v^2)/2 \tag{12}$$

The surplus enjoyed by the representative online gamer corresponds to

$$CS_d = \int_0^f x \, dx \Leftrightarrow CS_d = \left[ x^2/2 \right]_0^f \Leftrightarrow f^2/2 \tag{13}$$

Consequently, social welfare is given by

$$SW = \mathrm{E}[\Pi] + CS_u + CS_d \tag{14}$$

The timing structure of the game is given as follows. In the first stage, the platform defines the price charged to viewers. In the second stage, the platform decides the linear compensation plan to be offered to the manager. In the third stage, the manager decides whether they accept the proposal or not. If the response is affirmative, then the manager chooses the respective level of effort. Finally, market participation occurs. The game is solved by the method of backward induction.

## 3. Analysis
### 3.1. Market Participation Stage

Following Katz and Shapiro [36], the platform forms rational expectations with respect to the adhesion of new members on both sides of the market, which are assumed to be fulfilled in equilibrium. Solving Equations (3) and (7) simultaneously, the number of agents that get on board for each side of the market is given by

$$n_u = (1 + e - p)/[1 - g\theta(1 - \gamma)] + (\varepsilon_u + \theta \, \varepsilon_d)/[1 - g\theta(1 - \gamma)] \tag{15}$$

$$n_d = [g(1 - \gamma)(1 + e - p)]/[1 - g\theta(1 - \gamma)] + [g(1 - \gamma)\varepsilon_u + \varepsilon_d]/[1 - g\theta(1 - \gamma)] \tag{16}$$

Both outcomes depend on prices, managerial effort, and uncertainty components. Moreover, both cross-group network externalities have a positive effect on membership.

### 3.2. Managerial Effort Stage

Before market participation occurs, the manager chooses the effort level to be spent on the attraction of new members by maximizing Equation (10). After substituting Equation (8) in Equation (10), knowing that $\varepsilon_u \sim N(0, \sigma_u^2)$ and $\varepsilon_d \sim N(0, \sigma_d^2)$, and relying on the statistical properties $\mathrm{E}(a + bX) = a + b\mathrm{E}(X)$ and $\mathrm{Var}(a + bX) = b^2 \mathrm{Var}(X)$, it follows

$$U_{CE} = \alpha_0 + \frac{p\alpha_1(1 + e - p)}{1 - g\theta(1 - \gamma)} - \frac{\rho \alpha_1^2 p^2 (\theta^2 \sigma_d^2 + \sigma_u^2)}{2[1 - g\theta(1 - \gamma)]^2} - \frac{e^2}{2} \tag{17}$$

Differentiating Equation (17) with respect to $e$ implies an effort level given by

$$e = \frac{p\alpha_1}{1 - g\theta(1 - \gamma)} \tag{18}$$

with $\partial^2 U_{CE}/\partial e^2 < 0$. For a given price $p$ charged to viewers and commission rate $\alpha_1$ offered by the platform, the manager has an incentive to engage in effort when the intensity of both indirect network effects increases since $\partial e/\partial \theta > 0$ and $\partial e/\partial g > 0$ holds. Moreover, the effort level can be rewritten as

$$e = \kappa_1 \alpha_1 p \tag{19}$$

where $\kappa_1 := 1/[1 - g\theta(1 - \gamma)] > 1$, $\forall g > 0, \theta > 0, 0 < \gamma < 1$. This parameter corresponds to a multiplier effect of the access charge scheme applied to both sides of the market (i.e., directly on viewers through the price $p$, whereas indirectly on online gamers via the commission rate $\alpha_1$) on the managerial effort, which turns out to be positively influenced by both indirect network effects. Consequently, both cross-group network externalities not only provide a direct financial reward to the manager based on membership attraction but also ensure a positive spillover effect due to the feedback loop of viewers on online gamers

and vice-versa, which indirectly enhances the managerial valuation of marginal users. This multiplier, which stems from the specific nature of indirect network effects, is equivalent to say that the effort level spent by the manager on membership attraction is leveraged while keeping constant the number of new members that get on board. After substituting Equation (18) in Equation (17), the certainty equivalent is given by

$$U_{CE} = \alpha_0 + \frac{\alpha_1 p}{2[1 - g\theta(1 - \gamma)]^2} \left\{ 2(1 - p)[1 - g\theta(1 - \gamma)] + \alpha_1 p\left[1 - \rho\left(\theta^2\sigma_d^2 + \sigma_u^2\right)\right] \right\} \quad (20)$$

Doing similar procedure for the expected profit $\mathrm{E}[\Pi] = p(1 - \alpha_1)\,\mathrm{E}[n_u] + \gamma g\,\mathrm{E}[n_d] - \alpha_0$ secured by the platform and knowing that rational expectations are fulfilled in equilibrium, one obtains that the intermediation profit is given by

$$\mathrm{E}[\Pi] := \Pi = -\alpha_0 + \frac{[p(1 - \alpha_1) + g^2\gamma(1 - \gamma)][1 - p(1 - \alpha_1) - g\theta(1 - \gamma)(1 - p)]}{[1 - g\theta(1 - \gamma)]^2} \quad (21)$$

*3.3. Compensation Plan Stage*

3.3.1. Market Environment Characterized by Indirect Network Effects

Before market participation and effort level decisions, one must ensure that the certainty equivalent is null, which is equivalent to say that the fixed salary $\alpha_0$ is defined in such a way to make the IR constraint binding due to the profit-maximizing behavior adopted by the platform. Using Equation (20), it is clear that

$$U_{CE} = 0 \Leftrightarrow \alpha_0 = -\frac{\alpha_1 p}{2[1 - g\theta(1 - \gamma)]^2} \left\{ 2(1 - p)[1 - g\theta(1 - \gamma)] + \alpha_1 p\left[1 - \rho\left(\theta^2\sigma_d^2 + \sigma_u^2\right)\right] \right\} \quad (22)$$

Substituting this threshold in Equation (21) implies an intermediation profit given by

$$\begin{aligned} \Pi \;=\; & \frac{2(1 - p)[1 - g\theta(1 - \gamma)][p + g^2\gamma(1 - \gamma)]}{2[1 - g\theta(1 - \gamma)]^2} \\ & + \frac{\alpha_1 p}{2[1 - g\theta(1 - \gamma)]^2} \left\{ 2[p + g^2\gamma(1 - \gamma)] - \alpha_1 p[1 + \rho(\theta^2\sigma_d^2 + \sigma_u^2)] \right\} \end{aligned} \quad (23)$$

The platform ensures profit maximization by differentiating Equation (23) with respect to $\alpha_1$, which implies a commission rate given by

$$\alpha_1 = \frac{\kappa_2}{1 + \rho\left(\theta^2\sigma_d^2 + \sigma_u^2\right)} \quad (24)$$

with $\kappa_2 := 1 + g^2\gamma(1 - \gamma)/p > 1$, $\forall g > 0$, $p > 0$, $0 < \gamma < 1$. The second-order condition (SOC) given by $\partial^2\Pi/\partial\alpha_1^2 = -p^2[1 + \rho(\theta^2\sigma_d^2 + \sigma_u^2)]/[1 - g\theta(1 - \gamma)]^2$ is unambiguously satisfied.

**Corollary 1.** *The commission rate $\alpha_1$ is inversely related to the cross-group network externality $\theta$ exerted on viewers, but positively influenced by the cross-group network externality $g$ exerted on online gamers.*

**Proof.** Supplementary Material. □

Corollary 1 is justified by the one-sided revenue-based managerial contract scheme, where the platform knows that the manager is supposed to engage in an ex-post effort to attract viewers. A profit-maximizing behavior means that the monopoly-holding platform has the incentive to increase the managerial compensation when viewers have the ability to attract additional online gamers ($\partial\alpha_1/\partial g > 0$), while the opposite is expected to hold when online gamers have the ability to attract viewers ($\partial\alpha_1/\partial\theta < 0$). Although not making any explicit assumption on this particular domain, this explanation suggests that the strongest (weakest) side of the market from the platform's perspective in terms of membership attraction is the group composed of online gamers (viewers), respectively. This argument

ultimately justifies the reason behind the need to hire a manager: the strongest side is easily attracted to the platform without the need for third-party assistance, while the specialized support has a mandatory nature to ensure that agents from the weakest side of the market get on board. Furthermore, one should emphasize that the previous explanation is not necessarily at odds with the managerial vision according to which the strongest (weakest) side of the market in terms of membership attraction is the group composed of viewers (online gamers), respectively. As the reader can realize by checking the content of the following Subsection, viewers ensure a direct compensation to the manager based on Equation (8), while online gamers have the ability to attract agents from the opposite side of the market by themselves, thus, indirectly harming the compensation obtained by the manager. In fact, Corollary 1 shows that $\partial \alpha_1 / \partial \theta < 0$, while one can observe that $\partial e / \partial \theta < 0$ holds in the subsequent stage by differentiating Equation (18) with respect to the relevant parameter. However, Corollary 1 reveals that $\partial \alpha_1 / \partial g > 0$, which coincides with the observation that $\partial e / \partial g > 0$ after differentiating Equation (18) with respect to the relevant parameter. This implies that the compensation plan defined by the principal and the effort spent by the agent respond differently (similarly) to the indirect network effect that affects viewers (online gamers), respectively.

### 3.3.2. Comparison of Effectiveness and Risk with Markets Absent of Indirect Network Effects

Effectiveness corresponds to the effect resulting from a marginal increment of the commission rate $\alpha_1$ on the effort employed by the manager to bring new viewers on board, while holding everything else constant. Although effectiveness is given by the unit value in markets absent of network effects, it becomes clear after rearranging Equation (18) that effectiveness corresponds to the parameter $\kappa_1$ in two-sided markets. Risk is a concept that refers to the membership (or, alternatively, sales) variance faced by the platform. While risk is given by the variance of membership $\sigma^2$ in markets absent of network effects, the respective value is dependent on $\sigma_d^2$ and $\sigma_u^2$ in two-sided markets. In a market absent of indirect network effects, the risk is assessed by the formula

$$\text{Risk} = \left( \frac{1 - \alpha_1}{\rho \, \alpha_1} \right) \text{Effectiveness}^2 \tag{25}$$

where the formal association between effectiveness, the commission rate $\alpha_1$, degree of risk aversion $\rho$, and risk faced by members is observed. Note that the commission rate without network externalities corresponds to $\alpha_1 = 1 / (1 + \rho \sigma^2)$, which is consistent with the equality identified in Equation (25). However, differently from markets characterized by the absence of indirect network effects where effectiveness and risk are independent of each other, cross-group network externalities interconnect both concepts in two-sided markets given that the optimal balance between effectiveness and risk achievable by the principal in the moment of deciding about the commission rate is expected to be affected by indirect network effects. Although seminal studies in the salesforce compensation literature indicate that the commission rate increases as effectiveness (risk) increases (decreases) in markets absent of cross-group network externalities [16], a complementary view reveals a differentiated behavior between effectiveness and risk in the context of two-sided markets, respectively. Table S1 in Supplementary Material compiles effectiveness and risk values, which confirm that indirect network externalities have an asymmetric impact on both concepts.

Knowing that risk consists of components $\sigma_d^2$ and $\sigma_u^2$ in two-sided markets, the main differences relative to a market environment absent of indirect network effects are twofold. First, the platform internalizes that the risk is partially bypassed from online gamers to viewers when agents from the former group attract agents from the latter one due to the cross-group network externality $\theta$ (i.e., the effect $\theta^2 \sigma_d^2$ corresponding to a membership variance term). Second, as already suggested by Corollary 1, the allocative distortion of risk is not fully compensated by effectiveness gains, which is equivalent to say that the principal is forced to set a commission rate inversely proportional to $\theta$. In other words, the

commission rate provided by the platform to the manager decreases in the magnitude of viewers being attracted to the platform by online gamers since this mechanism alleviates the need for the manager to exert additional effort to attract viewers. Consequently, the proactive action developed by the online gamer's side harms the compensation scheme enjoyed by the manager.

### 3.3.3. Remaining Outcomes and the Role of Cross-Group Network Externalities on Profit

Substituting Equation (24) in the remaining endogenous variables, one obtains

$$e = \frac{p + g^2\gamma(1-\gamma)}{[1 - g\theta(1-\gamma)][1 + \rho(\theta^2\sigma_d^2 + \sigma_u^2)]} \tag{26}$$

$$n_u = \frac{1-p}{[1 - g\theta(1-\gamma)]} + \frac{p + g^2\gamma(1-\gamma)}{[1 - g\theta(1-\gamma)]^2[1 + \rho(\theta^2\sigma_d^2 + \sigma_u^2)]} \tag{27}$$

$$n_d = \frac{g(1-\gamma)[1 - p + g\theta(1-p)(1-\gamma)]}{[1 - g\theta(1-\gamma)]^2} + \frac{g(1-\gamma)[p + g^2\gamma(1-\gamma)]}{[1 - g\theta(1-\gamma)]^2[1 + \rho(\theta^2\sigma_d^2 + \sigma_u^2)]} \tag{28}$$

$$CS_u = \frac{1}{2} - \frac{\left\{[1 - g\theta(1-\gamma)][p + g\theta(1-\gamma)] - \frac{[p+g^2\gamma(1-\gamma)]}{[1+\rho(\theta^2\sigma_d^2+\sigma_u^2)]}\right\}^2}{2[1 - g\theta(1-\gamma)]^4} \tag{29}$$

$$CS_d = \frac{g^2(1-\gamma)^2\left[1 - p + g\theta(1-p)(1-\gamma) + \frac{p+g^2\gamma(1-\gamma)}{1+\rho(\theta^2\sigma_d^2+\sigma_u^2)}\right]^2}{2[1 - g\theta(1-\gamma)]^4} \tag{30}$$

According to the theory of two-sided markets focused on monopoly market structures, indirect network effects increase the utility of agents from both sides of the market [13]. This action makes the platform more attractive to both types of users, which enhances the intermediation profit. In light of this study, the profit of the platform can be divided into two parts where one component corresponds to the market contribution ($\Lambda_1$), while the other corresponds to the managerial contribution ($\Lambda_2$). Formally, $\Pi = \Lambda_1 + \Lambda_2$ is given by

$$\Pi = \frac{(1-p)[p + g^2\gamma(1-\gamma)]}{1 - g\theta(1-\gamma)} + \frac{[p + g^2\gamma(1-\gamma)]^2}{2[1 - g\theta(1-\gamma)]^2[1 + \rho(\theta^2\sigma_d^2 + \sigma_u^2)]} \tag{31}$$

**Corollary 2.** *Intermediation profit is influenced by indirect network effects θ and g as follows.*

*(I)   For each individual component:*

- $\Lambda_1$ *is positively affected by both cross-group network externalities (i.e., $\partial\Lambda_1/\partial\theta > 0$ $\cap$ $\partial\Lambda_1/\partial g > 0$).*
- $\Lambda_2$ *is positively influenced by the cross-group network externality on online gamers (i.e., $\partial\Lambda_2/\partial g > 0$), while being either positively or negatively influenced by the cross-group network externality affecting viewers according to the following rule of inequalities*

$$\partial\Lambda_2/\partial\theta > (\leq) \, 0 \Leftrightarrow g(1-\gamma)\left(1 + \rho\sigma_u^2\right) > (\leq) \, \rho\theta\sigma_d^2[1 - 2g\theta(1-\gamma)]$$

*(II)   In aggregate terms:*

- *Profit is unambiguously influenced by the cross-group network externality exerted on online gamers (i.e., $\partial\Pi/\partial g > 0$).*
- *Profit can be either positively or negatively affected by the cross-group network externality exerted on viewers (i.e., $\partial\Pi/\partial\theta > (\leq) \, 0$ holds).*

**Proof.** Supplementary Material. □

After disaggregating the profit enjoyed by the platform, one observes that the component associated with the managerial mechanism on membership attraction can partially distort the result yielding under perfect information. As such, the main message behind Corollary 2 is that introducing the principal-agent model with demand uncertainty on both sides of the market does not imply that the platform is always capable of leveraging profit for a stronger intensity of the indirect network externality exerted by online gamers on viewers. The intuition behind this result is straightforward. When viewers have a higher ability to attract online gamers or, similarly, when the valuation of one additional viewer increases for online gamers (i.e., higher $g$), these have a higher incentive to attract agents from the opposite side of the market to secure a higher surplus. This is a rent-seeking action for the platform since the need to pay an excessively high bonus to the manager is reduced. In turn, when online gamers have a higher ability to attract viewers or, similarly, when the valuation of one additional online gamer increases for viewers (i.e., higher $\theta$), these exhibit a strong desire to consume additional content. Since online gamers have an incentive to perform this kind of action to secure additional revenue, the platform optimally reacts by providing a higher compensation to the manager in order to ensure that additional viewers join the platform. Therefore, this novel result is justified by the fact that the cross-group network externality on online gamers (viewers) corresponds to a spillover effect owned by the intermediary, which has the ability to bypass demand uncertainty related to viewers (online gamers) to the imposition of managerial incentives in such a way that the commission rate decreases (increases) and profit increases (decreases), respectively.

### 3.4. Viewers Pricing Stage

The initial task of the platform is to decide the access charge applied to viewers. Profit maximization requires differentiating Equation (31) with respect to $p$, which implies

$$p = \frac{1 - g^2\gamma(1-\gamma)}{2} + \frac{1 + g^2\gamma(1-\gamma)}{2\{1 - 2g\theta(1-\gamma) + 2\rho[1 - g\theta(1-\gamma)](\theta^2\sigma_d^2 + \sigma_u^2)\}} \quad (32)$$

## 4. Results

In what follows, let us clarify the main results of this study.

### 4.1. Subgame Perfect Nash Equilibrium

**Proposition 1.** *Let $0 < \rho \leq \bar{\rho}$, $0 < \sigma_u \leq \bar{\sigma}_u$ and $0 < g < \bar{g}$ be parameters dependent on viewers. For any $\sigma_d > 0$, $\theta > 0$, and $0 < \gamma < 1$ driven by online gamers, each stage of the SPNE is characterized by the following equilibrium outcomes.*

$$
\begin{cases}
p^* = \frac{1 - g^2\gamma(1-\gamma)}{2} + \frac{1 + g^2\gamma(1-\gamma)}{2\{1 - 2g\theta(1-\gamma) + 2\rho[1 - g\theta(1-\gamma)](\theta^2\sigma_d^2 + \sigma_u^2)\}} \\[2mm]
\alpha_1^* = \frac{[1 + g^2\gamma(1-\gamma)][1 - g\theta(1-\gamma)]}{1 - [1 - g^2\gamma(1-\gamma)]\{g\theta(1-\gamma) + \rho[1 - g\theta(1-\gamma)](\theta^2\sigma_d^2 + \sigma_u^2)\}} \\[2mm]
e^* = \frac{1 + g^2\gamma(1-\gamma)}{1 - 2\{g\theta(1-\gamma) + \rho[1 - g\theta(1-\gamma)](\theta^2\sigma_d^2 + \sigma_u^2)\}} \\[2mm]
n_u^* = \frac{[1 + g^2\gamma(1-\gamma)][1 + \rho(\theta^2\sigma_d^2 + \sigma_u^2)]}{1 - 2\{g\theta(1-\gamma) + \rho[1 - g\theta(1-\gamma)](\theta^2\sigma_d^2 + \sigma_u^2)\}} \\[2mm]
n_d^* = \frac{g(1-\gamma)[1 + g^2\gamma(1-\gamma)][1 + \rho(\theta^2\sigma_d^2 + \sigma_u^2)]}{1 - 2\{g\theta(1-\gamma) + \rho[1 - g\theta(1-\gamma)](\theta^2\sigma_d^2 + \sigma_u^2)\}}
\end{cases}
$$

The remaining endogenous variables are given by

$$
\begin{cases}
CS_u^* = \frac{1}{2} - \dfrac{\left\{ g(1-\gamma)(g\gamma+2\theta) - \rho\left[1 - g(1-\gamma)(g\gamma+2\theta)\right]\left(\theta^2\sigma_d^2+\sigma_u^2\right)\right\}^2}{2\left(1 - 2\left\{ g\theta(1-\gamma)+\rho\left[1-g\theta(1-\gamma)\right]\left(\theta^2\sigma_d^2+\sigma_u^2\right)\right\}\right)^2} \\[4mm]
CS_d^* = \dfrac{g^2(1-\gamma)^2\left[1+g^2\gamma(1-\gamma)\right]^2\left[1+\rho\left(\theta^2\sigma_d^2+\sigma_u^2\right)\right]^2}{2\left(1 - 2\left\{ g\theta(1-\gamma)+\rho\left[1-g\theta(1-\gamma)\right]\left(\theta^2\sigma_d^2+\sigma_u^2\right)\right\}\right)^2} \\[4mm]
\Pi^* = \dfrac{\left[1+g^2\gamma(1-\gamma)\right]^2\left[1+\rho\left(\theta^2\sigma_d^2+\sigma_u^2\right)\right]}{2\left\{ 1-2g\theta(1-\gamma)+2\rho\left[1-g\theta(1-\gamma)\right]\left(\theta^2\sigma_d^2+\sigma_u^2\right)\right\}}
\end{cases}
$$

Equilibrium social welfare corresponds to their sum, with

$$
\overline{\rho} := \frac{g\theta(1-\gamma)\left[1-g^2\gamma(1-\gamma)\right]-1}{\left[1-g^2\gamma(1-\gamma)\right]\left[1-g\theta(1-\gamma)\right]\left(\theta^2\sigma_d^2+\sigma_u^2\right)}
$$

$$
\overline{\sigma}_u := \sqrt{\frac{g(1-\gamma)(g\gamma+2\theta)-\rho\theta^2\sigma_d^2\left[1-g(1-\gamma)(g\gamma+2\theta)\right]}{\rho\left[1-g(1-\gamma)(g\gamma+2\theta)\right]}}
$$

$$
\overline{g} := \frac{1+2\rho\left(\theta^2\sigma_d^2+\sigma_u^2\right)}{2\theta(1-\gamma)\left[1+\rho\left(\theta^2\sigma_d^2+\sigma_u^2\right)\right]}
$$

**Proof.** Supplementary Material. □

*4.2. Impact of Cross-Group Network Externalities on the Viewer's Price*

**Lemma 1.** *Let* $0 < \rho \le \overline{\rho}$, $0 < \sigma_u \le \overline{\sigma}_u$, $0 < g < \overline{g}$, $\sigma_d > 0$, $\theta > 0$ *and* $0 < \gamma < 1$.

*(I)*   *For the cross-group network externality $\theta$ exerted by online gamers on viewers:*

$$
\frac{\partial p^*}{\partial \theta} \le 0 \text{ iff } g \in (0,\widetilde{g}] \;\cup\; g \in (\widetilde{g},\overline{g}] \,\cap\, \gamma \in [\,\widetilde{\gamma},1)
$$

*otherwise* $\partial p^*/\partial\theta > 0$ *holds.*

*(II)*   *For the cross-group network externality  g exerted by viewers on online gamers:*

$$
\frac{\partial p^*}{\partial g} \ge 0 \text{ iff } \rho \in (0,\widetilde{\rho}]
$$

*otherwise* $\partial p^*/\partial g < 0$ *holds, with*

$$
\widetilde{g} := \frac{2\rho\theta\sigma_d^2}{1+\rho\left(3\theta^2\sigma_d^2+\sigma_u^2\right)}
$$

$$
\widetilde{\gamma} := 1 - \frac{2\rho\theta\sigma_d^2}{g\left[1+\rho\left(3\theta^2\sigma_d^2+\sigma_u^2\right)\right]}
$$

$$
\widetilde{\rho} := \frac{\sqrt{4g^2\gamma^2+4g\gamma\theta\left[3+g^2\gamma(1-\gamma)\right]+\theta^2\left\{1-g^2\gamma(1-\gamma)\left[10+7g^2\gamma(1-\gamma)\right]\right\}}}{8g\gamma\left[1+g\theta(1-\gamma)\right]^2\left(\theta^2\sigma_d^2+\sigma_u^2\right)^2} \\
+ \frac{\theta-g\gamma\left\{2-g\theta(1-\gamma)\left[11-8g\theta(1-\gamma)\right]\right\}}{8g\gamma\left[1+g\theta(1-\gamma)\right]^2\left(\theta^2\sigma_d^2+\sigma_u^2\right)^2}
$$

**Proof.** Supplementary Material. □

Knowing that the platform operates in a monopoly environment and deals with uncertainty at the membership level in both sides of the market, the first point of Lemma 1 confirms that a stronger intensity of the cross-group network externality $\theta$ fostered by an

additional online gamer on viewers has a positive effect on the price charged to viewers when two cumulative conditions are met: on the one hand, when online gamers are extremely attracted by viewers because the benefit $g$ of an additional viewer on online gamers is sufficiently high and, on the other hand, when the royalty rate paid by online gamers to the platform is too low such that the intermediary compensates the lack of revenue from the group of online gamers by charging a higher price to the opposite side of the market. Differently from Armstrong [13], a stronger intensity of the cross-group network externality $\theta$ fostered by an additional online gamer on viewers has a negative effect on the price charged to viewers either when online gamers are not excessively attracted by viewers or when the degree of attraction is considerably strong, but the royalty rate applied to online gamers is sufficiently high such that the platform can smooth the access charge on viewers.

While holding everything else constant, the second point of Lemma 1 indicates that a stronger intensity of the cross-group network externality $g$ fostered by an additional viewer on online gamers has a positive (negative) effect on the price applied to viewers as long as the degree of risk aversion affecting viewers is sufficiently low (high), respectively. The intuition behind this result is straightforward. As viewers become more attractive in the eyes of online gamers, the platform has an incentive to increase (decrease) the price applied to viewers as long as this exhibits a high (low) propensity to consume content created by online gamers, respectively.

*4.3. Impact of the Asymmetric Gap between Indirect Network Effects on the Compensation Plan*

Similar to Ribeiro [15], let us take as a given the presence of a gap in the indirect network effect sustained by each side of the market represented by $d$, with $0 \leq d \leq 1$. Assume $\theta := g(1-d)$, so that the benefit of an additional online gamer to viewers cannot be greater than the benefit of an additional viewer to online gamers. One expects that parameter $\theta$ is never higher than parameter $g$ since only this consideration justifies gamers' action of attracting additional viewers to the platform, which reduces the platform's incentive of providing a sufficiently high bonus to the manager under the condition described in Corollary 2. We consider that the asymmetric gap between indirect network effects increases in $d$ given that, under the limit case $d = 1$ ($d = 0$), the benefit of one additional online gamer to viewers is null for any given (equal to the) benefit that one additional viewer provides to online gamers, respectively. This refinement allows us to assess the impact of a change in parameter $d$ on the commission rate of the incentive scheme provided by the platform to the manager.

**Lemma 2.** *Let* $0 < \rho \leq \bar{\rho}$, $0 < \sigma_u \leq \bar{\sigma}_u$, $0 < g < \bar{g}$, $\sigma_d > 0$, $\theta > 0$, $0 < \gamma < 1$ *and consider the gap between indirect network effects* $0 \leq d \leq 1$ *such that* $\theta := g(1-d)$.

*(I)*     *In the absence of bilateral demand uncertainty:*

$$\left. \frac{\partial \alpha_1^*}{\partial d} \right|_{(\sigma_u, \sigma_d) \to (0,0)} > 0$$

*(II)*    *Under the presence of uncertainty in the number of active viewers:*

$$\left. \frac{\partial^2 \alpha_1^*}{\partial \sigma_u \partial d} \right|_{\sigma_d \to 0} > 0$$

*(III)*   *Under the presence of uncertainty in the number of active online gamers:*

$$\left. \frac{\partial^2 \alpha_1^*}{\partial \sigma_d \partial d} \right|_{\sigma_u \to 0} > 0$$

**Proof.** Supplementary Material. □

The first point of Lemma 2 demonstrates that an increasing gap between indirect network effects has a strictly positive impact on the variable managerial bonus provided by the monopoly-holding platform. This result complements the conclusion applied to competitive markets detailed in Ribeiro [15], where the variable managerial bonus is inversely related to the gap between indirect network effects. Taking the increasing gap between indirect network effects as a given, the remaining points of Lemma 2 show that, the higher the intensity of demand uncertainty on membership, the greater the variable compensation absorbed by the manager is expected to be. Intuitively, this result is justified by the platform's higher necessity to have managerial support to get additional agents from both sides of the market on board for increasing dilution of perfect information with respect to their adhesion.

*4.4. Managerial Reaction to the Asymmetric Gap between Indirect Network Effects*

**Lemma 3.** *Let* $0 < \rho \leq \bar{\rho}$, $0 < \sigma_u \leq \bar{\sigma}_u$, $0 < g < \bar{g}$, $\sigma_d > 0$, $\theta > 0$, $0 < \gamma < 1$ *and consider the gap between indirect network effects* $0 \leq d \leq 1$ *such that* $\theta := g(1 - d)$. *Then*

$$\frac{\partial e^*}{\partial d} > 0 \text{ iff } \sigma_d^2 > \widetilde{\sigma}_d \cap \rho \in (\hat{\rho}, \bar{\rho}]$$

Otherwise $\partial e^* / \partial d \leq 0$ holds, with

$$\widetilde{\sigma}_d := \frac{g^2 \gamma \sigma_u^2 (1 - \gamma)^2}{(1 - d)\left\{2 - g^2(1 - d)(1 - \gamma)\left[4 - g^2(2 - 2d + \gamma)(1 - \gamma) + 2g^4 \gamma(1 - d)(1 - \gamma)^2\right]\right\}}$$

$$\hat{\rho} := \frac{1 - \gamma}{\sigma_d^2(1 - d)[2 - 3g^2(1 - d)(1 - \gamma)] - \sigma_u^2(1 - \gamma)}$$

**Proof.** Supplementary Material. □

Lemma 3 indicates that the managerial effort is positively influenced by the asymmetric gap between indirect network externalities (i.e., when viewers become increasingly more valuable than online gamers or, equivalently, when online gamers become increasingly less valuable than viewers) when two cumulative conditions are verified in equilibrium: on the one hand, when the uncertainty related to the membership of online gamers is too high and, on the other hand, when the risk aversion of viewers on joining the online gaming industry is excessive.

Recall that the revenue-based contract signed between principal and agent is one-sided. Moreover, Corollary 2 confirms market conditions where the platform has a lower incentive to provide generous compensation to the manager since viewers can join the platform based on actions developed by online gamers. In light of the mathematical formalization adopted in this study, an increasing asymmetric gap between indirect network effects implies that viewers (online gamers) are more (less) important than online gamers (viewers) because an additional agent from this side that becomes actively engaged in the market provides a higher (lower) benefit to the counterpart, respectively.

Given the persistence of a market asymmetry in favor of the overvaluation of viewers, the restriction imposed on $\sigma_d$ reflects that the manager has a higher incentive to spend additional effort on attracting viewers the more uninformed the platform is about the number of online gamers that get on board.

Moreover, the restriction imposed on $\rho$ means that highly risk-averse viewers with respect to online gaming activities give to the manager a stronger incentive to spend additional effort on attracting viewers. While the higher managerial effort under the

first type of restriction is caused by a market failure related to the absence of credible information to serve the interest of the platform, the higher managerial effort under the second type of restriction is justified by the reluctance of adhesion to the platform by the most valuable side of the market from the manager's perspective.

*4.5. Market Shares, Cross-Group Network Externalities, and Membership Uncertainty*

**Corollary 3.** *Let* $0 < \rho \leq \bar{\rho}$, $0 < \sigma_u \leq \bar{\sigma}_u$, $0 < g < \bar{g}$, $\sigma_d > 0$, $\theta > 0$ *and* $0 < \gamma < 1$. *Inequality* $n_u^* > n_d^*$ *unambiguously holds in equilibrium.*

**Proof.** Supplementary Material. □

Corollary 3 confirms that there are more viewers than online gamers in equilibrium. One should also highlight that those inequalities $\partial n_u^*/\partial \rho < 0$, $\partial n_u^*/\partial \sigma_u < 0$, and $\partial n_d^*/\partial \sigma_d < 0$ hold in equilibrium. The first one reveals that a lower number of viewers stays active in the market as the risk aversion faced by viewers converges to the peak value $\bar{\rho}$. The remaining inequalities clarify that within-side demand uncertainty leads to a reduction of the number of viewers and online gamers that get on board. The effect of indirect network effects and demand uncertainty on the equilibrium number of agents on board on the opposite side of the market is described as follows.

**Lemma 4.** *Let* $0 < \rho \leq \bar{\rho}$, $0 < \sigma_u \leq \bar{\sigma}_u$, $0 < g < \bar{g}$, $\sigma_d > 0$, $\theta > 0$ *and* $0 < \gamma < 1$.
*(I)     On the side of viewers:*

$$\frac{\partial n_u^*}{\partial \sigma_d} < 0 \, , \, \frac{\partial n_u^*}{\partial g} > 0$$

*(II)    On the side of online gamers:*

$$\frac{\partial n_d^*}{\partial \rho} < 0 \, , \, \frac{\partial n_d^*}{\partial \sigma_u} < 0, \text{ ambiguity in } \frac{\partial n_d^*}{\partial \theta}$$

**Proof.** Supplementary Material. □

Two important conclusions are revealed. On the one hand, Lemma 4 shows that a higher degree of demand uncertainty faced by the platform with respect to the adhesion of online gamers decreases the equilibrium number of active viewers. Moreover, a higher benefit promoted by an additional viewer on online gamers increases the equilibrium number of active viewers. On the other hand, Lemma 4 shows that a higher degree of demand uncertainty faced by the platform with respect to the adhesion of viewers decreases the equilibrium number of online gamers that get on board. Moreover, a higher risk aversion faced by viewers discourages the presence of additional active online gamers.

Overall, these results suggest that efforts to reduce either information asymmetry or market inertia (e.g., risk aversion of viewers) are clearly advised to boost the market participation of agents. In turn, the variability of $n_d^*$ for a unit change in $\theta$ is more complex. In the absence of bilateral demand uncertainty, a stronger intensity of the cross-group network externality $\theta$ affecting viewers has a positive impact on the equilibrium number of online gamers ($\partial n_d^*/\partial \theta|_{(\sigma_u, \sigma_d) \to (0,0)} > 0$). Similar results are applied to the case where demand uncertainty prevails only in viewers given that a stronger intensity of the cross-group network externality $\theta$ affecting viewers has a positive impact on the equilibrium number of online gamers ($\partial n_d^*/\partial \theta|_{\sigma_u \to 0} > 0$). However, when assessing the sign of the derivative under the presence of bilateral demand uncertainty, one observes the persistence of ambiguous impact on the equilibrium number of active viewers for a marginal increase in the indirect network effect $g$. Knowing that analytical computations are untreatable,

several simulations were developed to meet this purpose. Figure 1 exposes the simulation outcomes, which allow us to provide some intuition on the comparative statics.

Panels A, B, and D show that $\partial n_d^* / \partial \theta$ increases (decreases) in $\theta$ if parameters $\sigma_d$, $\gamma$ and $\rho$ are sufficiently low (high), respectively. Economically speaking, the fact that online gamers have a greater ability to attract viewers leads to a rise (reduction) of market participation in the group of online gamers as long as the demand uncertainty, royalty rate charged by the platform, and risk aversion faced by viewers exhibit a sufficiently low (high) value, respectively.

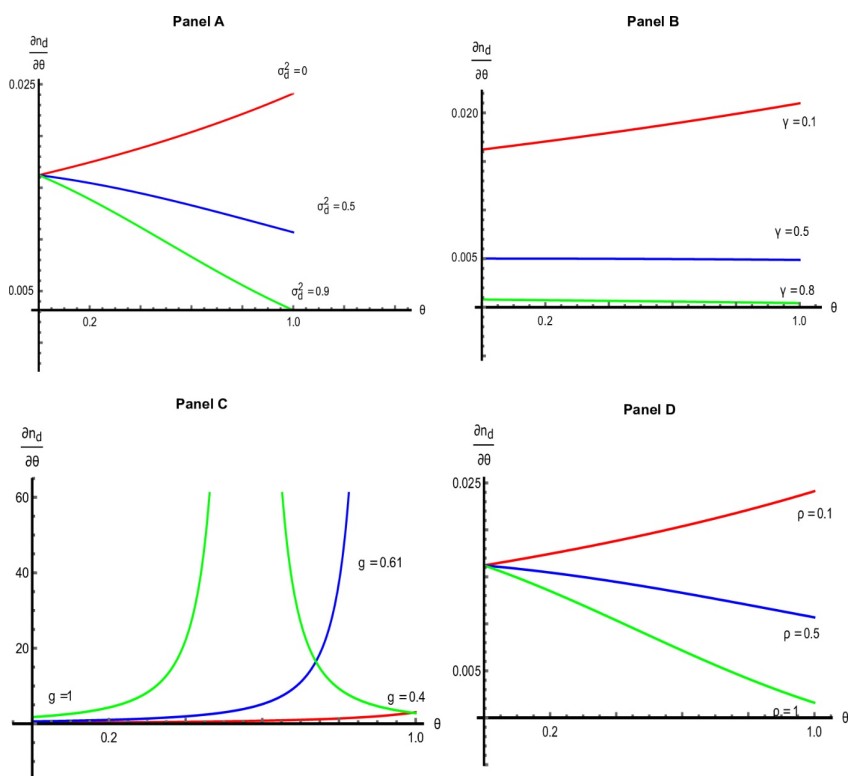

**Figure 1.** Impact of the indirect network effect affecting viewers on the equilibrium number of online gamers for a given change in parameters $\sigma_d^2$, $\gamma$, $g$, and $\rho$. Panel A simulates the variation of $\partial n_d^* / \partial \theta$ for different values taken by parameter $\sigma_d^2$. Panel B simulates the variation of $\partial n_d^* / \partial \theta$ for different values taken by parameter $\gamma$. Panel C simulates the variation of $\partial n_d^* / \partial \theta$ for different values taken by parameter $g$. Panel D simulates the variation of $\partial n_d^* / \partial \theta$ for different values taken by parameter $\rho$.

Panel C also shows that $\partial n_d^* / \partial \theta$ increases (decreases) in $\theta$ if $g$ is sufficiently low (high), respectively. However, the main difference compared to other panels is that the reduction of $\partial n_d^* / \partial \theta$ in $\theta$ for a sufficiently high $g$ corresponds to a non-monotone relation rather than evolving linearly in $\theta$. This is because panel C shows that the derivative under evaluation only decreases for a sufficiently strong intensity of the indirect network effect $\theta$ affecting viewers as long as the intensity of the indirect network effect $g$ affecting online gamers is also sufficiently strong. Differently from Armstrong [13] but similar to the equilibrium price applied to viewers, we conclude that it is not necessarily true that a stronger cross-group network externality on viewers always increases the equilibrium number of online gamers on board in markets controlled by a monopoly-holding platform.

## 5. Surplus Enjoyed by Each Side of the Market and Intermediation Profit

*5.1. Theoretical Outcome*

**Lemma 5.** *Let $0 < \rho \leq \bar{\rho}$, $0 < \sigma_u \leq \bar{\sigma}_u$, $0 < g < \bar{g}$, $\sigma_d > 0$, $\theta > 0$ and $0 < \gamma < 1$.*

(I)    The impact of $\theta$ on $CS_u^*$, ($CS_d^*$ and $\Pi^*$) is ambiguous, but opposite (equivalent) in terms of sign to that holding for $n_d^*$, respectively.

(II)   The impact of $g$ on $CS_u^*$, $CS_d^*$ and $\Pi^*$ is strictly positive.

**Proof.** Supplementary Material. □

Recall that the surplus enjoyed by online gamers is formally given by $CS_d = f^2/2$, so that the impact of a parameter variation on $f$ exhibits the same sign relatively to the impact of the same parameter on $CS_d$. The critical value of $f$ in equilibrium is given by

$$f_{crit} = n_d^* = \frac{g(1-\gamma)\left[1 + g^2\gamma(1-\gamma)\right]\left[1 + \rho\left(\theta^2\sigma_d^2 + \sigma_u^2\right)\right]}{1 - 2\left\{g\theta(1-\gamma) + \rho[1 - g\theta(1-\gamma)]\left(\theta^2\sigma_d^2 + \sigma_u^2\right)\right\}} \tag{33}$$

It follows that the findings applied to $n_d^*$ are qualitatively similar to those holding for $f_{crit}$. Hence, the sign of effects clarified in the second point of Lemma 4 and Figure 1 also holds for the equilibrium surplus enjoyed by online gamers. Furthermore, recall that the surplus enjoyed by viewers is given by $CS_u = (1 - v^2)/2$ such that the impact of a parameter change on $v$ has the opposite sign relative to the impact of the same parameter on $CS_u$. The critical value of $v$ in equilibrium is given by

$$v_{crit} = \frac{g(1-\gamma)(g\gamma + 2\theta) - \rho\left(\theta^2\sigma_d^2 + \sigma_u^2\right)\left[1 - g(1-\gamma)(g\gamma + 2\theta)\right]}{1 - 2\left\{g\theta(1-\gamma) + \rho[1 - g\theta(1-\gamma)]\left(\theta^2\sigma_d^2 + \sigma_u^2\right)\right\}} \tag{34}$$

Knowing that straightforward computations confirm that the numerator of $\partial v_{crit}/\partial\theta$ is qualitatively equivalent to that holding for $\partial n_d^*/\partial\theta$, we find that both $v_{crit}$ and $n_d^*$ suffer from similar effects for a given change in $\theta$. Consequently, the sign of the effect on $CS_u^*$, is necessarily opposed to that holding for $n_d^*$. Moreover, we confirm that $\partial v_{crit}/\partial g < 0$ such that $\partial CS_u^*/\partial g > 0$ is verified in equilibrium. Finally, differentiating the equilibrium profit described in Proposition 1 with respect to $\theta$ and $g$, one finds that the impact is qualitatively similar to that enjoyed by online gamers, which confirms that the interest of the platform is perfectly aligned with the interest of online gamers in the private equilibrium. This conclusion suggests that statistically significant determinants of the revenue enjoyed by online gamers are expected to have the same sign and influential power on the intermediation profit, which legitimizes the following empirical analysis. Finally, the result clarified in Corollary 2 is reinforced in Lemma 5 and, thus, the profit enjoyed by the monopoly-holding platform may decrease for increasing intensity of indirect network effects regardless of whether the price applied to viewers is endogenous or not.

*5.2. Empirical Validation*

Determinants of revenue enjoyed by an influential online gamer are assessed in the context of a monopoly environment (i.e., Twitch platform) characterized by uncertainty of membership on both sides of the market. Formally, this is equivalent to saying that we evaluate the impact of $\theta$ (i.e., the externality of the influential online gamer on the viewers' side) on $CS_d^*$ and, by direct effect due to the perfect alignment of interests, also on $\Pi^*$, while bearing in mind that the measurement of the cross-group network externality $\theta$ can take multiple forms (e.g., externality of the type 'being a follower', externality of the type 'being a subscriber', externality of the type 'being a viewer', etc.) as acutely explained below. This aspect is extremely relevant since it confirms that the idea that a unique type of indirect network effect is erroneous, and it does not have adherence to the observed reality in real-world platforms such as Twitch.

5.2.1. Data

Twitch is a video game live streaming service platform operated by Twitch Interactive, a subsidiary of Amazon. Introduced in June 2011 as a spin-off of the streaming platform Justin.tv, it primarily focuses on video game live streaming, including broadcasts of sports

competitions, music, games, creative content, and real-life streams. The platform provides a channel analytics page to online gamers, which allows them to obtain a comprehensive view of stream revenues and engagement statistics over customizable date ranges. These detailed breakdowns allow us to better understand the evolution of revenues and viewership trends. Several metrics of one influential Portuguese online gamer were monitored and collected during a period of 397 days predominantly covering the year 2017. Since most Portuguese online gamers use Twitch for streaming their activity and knowing that interests of the platform and online gamers are perfectly aligned based on Lemma 5, we believe that this empirical analysis is a good approximation for the monopoly environment that has been previously subject to theoretical treatment. Before moving into details about the selection of variables, one should clarify how the content produced by online gamers is rewarded at Twitch. While online gamers do not become Twitch affiliated members, they can obtain revenue only through donations directly sent by viewers to their PayPal account. After satisfying four entry criteria, online gamers can become Twitch affiliated members [37]. When that is the case, Twitch ensures that online gamers are able to monetize their channel by allowing them to benefit from additional sources of revenue that are subject to a 50-50 split with the platform.

Since the scope of analysis relies on revenues shared between online gamers and the platform, the dependent variable corresponds to revenues enjoyed by the online gamer with the exception of donations and commercial partnerships. These consist of a composite basket composed of channel subscription revenues and non-subscription revenues (information on subscription revenues can be segmented by type: paid subscriptions, Twitch Prime subscriptions, and gifted subscriptions. Information about non-subscription revenues can be segmented by source: ads, cheering (i.e., bits), game sales, extensions, bounties, and other bits interactions). As such, the empirical analysis considers three different models, which vary according to the type of dependent variable: model M1 considers only channel subscription revenues; model M2 considers only non-subscription revenues, and; model M3 considers total revenues. Knowing that each period of 24 h corresponds to a single observation in the dataset built to complete this empirical task, we consider as explanatory variables all the available statistics for online gamers at Twitch, which are summarized as follows.

Active data about the opposite side of the market (i.e., data on active actions conducted by viewers) include new followers (i.e., new followers received by the channel during live streaming in the selected date range), subscribers (i.e., number of subscribers in the selected date range), and live views (i.e., total views of live streams, which neither include video on demand (VOD) nor clip views).

Passive data about the opposite side of the market (i.e., data on passive actions conducted by viewers) include average viewers (i.e., the average number of concurrent viewers in a stream; to calculate this number, the platform identifies how many viewers there are at each point in time the online gamer is live streaming such that the final outcome is an average across all the time spent on live streaming in the selected date range), max viewers (i.e., the maximum number of viewers that the online gamer reached across all streams in the selected date range), unique viewers (i.e., the number of unique people who viewed the online gamer's live streams across the selected date range) and host/raid viewers (i.e., the percentage of viewers that come from hosts or raids (i.e., within-group externality between online gamers). Interactive data include time streamed (i.e., the total time of broadcasting in minutes), chat audience (i.e., the number of unique viewers who chatted across the selected date range), chat messages (i.e., the total number of chat messages sent), clips created (i.e., the number of clips created from streams), clip views (i.e., total views of clips created from streams), ad breaks (i.e., the total duration of ad breaks ran by the online gamer during streams in minutes), ad time per hour (i.e., the average amount of time per hour that ads were running during streams in minutes), notification engagements (i.e., the number of viewers engaged with go-live notifications sent out for streams in the selected data range). Data exclusively related to the online gamer's

characteristics include internet speed (i.e., the average download speed in the selected data range in Mbps) and psychological state of the online gamer (i.e., a dummy variable takes value 1 if the online gamer feels happy by the end of day $t-1$, while taking the value 0 otherwise).

Since the regressors have different units of measure, all values are standardized before performing the empirical analysis. This procedure maintains the anonymity of the online gamer intact and allows us to interpret the estimated coefficients as elasticities. For the sake of brevity, Table S2 in Supplementary Material compiles summary statistics.

### 5.2.2. Identification Strategy

Based on Lemma 5, we hypothesize that the revenue of the influential online gamer can either decrease or at least be subject to a negligible increment (i.e., approximately zero) for increasing intensity of the indirect network effect exerted on viewers. To understand whether this hypothesis holds in reality, three different machine learning models are trained and tested: principal component analysis (PCA), least absolute shrinkage and selection operator (LASSO), and a novel continual learning (CL) modeling approach based on the combination of random forest (RF) with ordinary least squares (OLS). All choices are justified by the high number of covariates and the subsequent need to mitigate concerns related to endogeneity, spurious correlation, omitted variable bias, and reverse causation.

For a regression estimator to meaningfully fit a model, it is mandatory the absence of omitted (i.e., confounding) variables correlate with covariates, the measurement of covariates should be done without error, covariates should not be correlated with the error term, and reverse causation should not occur (i.e., covariates affect the dependent variable, but not the opposite). Explanatory variables that do not satisfy these requirements are said to be endogenous. PCA is adopted to dissuade this concern since it corresponds to an unsupervised machine learning model that allows reducing the dimensionality of the initial set of covariates by creating a lower number of principal components (PC) that represent the initial set of covariates and ensure a proper evaluation of the dependent variable. The main advantages of this machine learning model include the provision of information about the relative contribution of each PC on explaining the total variance of a certain dependent variable, in addition to allowing for an economic interpretation of each extracted component to clearly express the respective content.

In turn, LASSO is a supervised machine learning model that uses a certain penalized regression technique to find the subset of variables from the initial set of covariates with significant explanatory power on the dependent variable. Despite avoiding concerns related to spurious correlation due to the reduction of dimensionality, many techniques exist to perform this operation. Knowing that coefficient estimates, and the set of independent variables depend on $\lambda$ (i.e., the general degree of penalization) and $\alpha$ (i.e., the relative contribution of $\ell 1$ versus $\ell 2$ norm penalization), a key question is how to choose tuning parameters. The most appropriate method depends on the setting and objective of the analysis, computational constraints, and if and how the independence and identically distribution assumption is violated. We use $k$-fold cross-validation and rolling $h$-step ahead cross-validation as penalized regularization techniques.

In a recent study, the authors of [38] developed a deep learning framework to ensure a more realistic learning analysis. According to the authors, CL is a new, simple, and efficient method proven to be valid as an alternative to standard regularization techniques. Common approaches to mitigate the omitted variable bias problem consider the execution of regularization to identify the relevant information that properly represents the past behavior of a dependent variable. While Ref. [38] adopted CL in the context of neural networks, we used this method in the broad context of machine learning. The idea of CL is to allow a refined treatment of covariates based on a bias-variance trade-off argument since it consists of a two-step approach that performs a bias-variance decomposition. In a first step, we apply RF to covariates with stronger relative importance on explaining the dependent variable in order to mitigate the risk of overfitting. We consider that

covariates that better explain the dependent variable are the active and passive data variables previously described. In a second step, we perform OLS estimation on predicted values obtained in the first step in order to mitigate the risk of underfitting. This option relative to other regularization techniques considers two important technical refinements: RF is initially applied to introduce higher bias and, thus, lower variance relative to OLS, which is a necessary condition to ensure generalization power; afterward, relying on predicted values obtained in the first step, OLS is applied to obtain non-biased regression estimates and, thus, higher variance in relation to RF (Bias (variance) is an error from erroneous assumptions in the learning algorithm (sensitivity to fluctuations in the training set). High bias (variance) can cause an algorithm to miss relations between features and target (model the random noise in training data rather than in the intended output), which fosters underfitting (overfitting), respectively).

### 5.2.3. Results

Tables S4–S6 in Supplementary Material compile estimated coefficients for each machine learning model. In what follows, results of each machine learning model are detailed.

### Principal Component Analysis

We apply the Kaiser's rule to conclude that each dependent variable is explained by 5 PCs (This rule states that the optimal number is given by PCs whose eigenvalue is above 1. Figure S1 in Supplementary Material shows a graphical representation of the final outcomes). Table S3 in Supplementary Material reveals that approximately 73% of the variance of each dependent variable is explained by these PCs. In terms of economic interpretation, knowing that PC1 is explained by the number of subscribers, followers, and lagged dependent variables, we conclude that it corresponds to a latent dimension that captures loyal viewers. PC2 is explained by unique viewers, streaming time, number of chat messages, and clip views, which suggests that it corresponds to a representative dimension of the non-faithful audience of the online gamer. PC3 corresponds to the publicity dimension since it is composed of covariates related to ads, while PC4 covers structural conditions faced by the online gamer to execute the activity (i.e., Internet speed and notifications capturing user engagement). PC5 is negatively (positively) affected by the percentage of host/raid viewers (chat audience and psychological state of the online gamer by the end of the previous day, respectively). Therefore, it consists of a latent component that captures the emotional dimension of the online gamer.

We then infer the following conclusions. First, the online gamer should be primarily focused on actions aimed at converting non-committed viewers into a fully committed audience by resorting to brand loyalty strategies. Second, this individual should make efforts to reduce the likelihood of being influenced by the emotional dimension since it appears to have a negative and significant effect on all types of revenue enjoyed by the online gamer. Nevertheless, this impact is particularly felt on the type of revenue that had the lowest incremental gain over time (i.e., non-subscription revenues). Third, statistically significant effects are resilient to different types of revenue, which suggests that this online gamer has the incentive to become professional. Finally, the hypothesis claiming that the online gamer's revenue can either decrease or at least be subject to a negligible increment for increasing intensity of the indirect network effect promoted on viewers cannot be rejected due to the negative and significant coefficient of PC2, which means that the negative impact on the gamer's revenue for increasing viewership is clearly promoted by the non-committed audience of the online gamer.

### Least Absolute Shrinkage and Selection Operator

Focusing on estimated coefficients with rolling h-step ahead cross-validation technique, a first conclusion is that covariates with explanatory power on the different types of dependent variables are the number of subscribers and live views. In the case of non-subscription revenues, ad breaks are also statistically significant. Nevertheless, one can

observe that only the number of subscribers has a considerably high magnitude on each dependent variable. Consequently, this result suggests that subscriptions have a positive impact on the different types of revenue enjoyed by the online gamer, but the remaining statistically significant covariates (i.e., followers and live views) have a negligible impact, particularly on non-subscription revenues. While the first conclusion is contrary to the null hypothesis claimed in the identification strategy, the second one is aligned with the idea that the revenue enjoyed by the online gamer is subject to a negligible gain for increasing intensity of the indirect network effect promoted on viewers. This regularization technique also allows us to obtain coefficients associated with a prediction for n days ahead. In addition to coefficients associated with one-day step-ahead prediction, we also consider those associated with 30 days step-ahead prediction, which allows us to conclude that the magnitude of estimated coefficients remains practically unchanged. Overall, the ambiguity of results yielding under the LASSO with rolling h-step ahead cross-validation technique demonstrates that PCA results are at least partially robust. Furthermore, the estimated coefficients under LASSO and post-estimation OLS are extremely similar, which reinforces the previous conclusion.

In turn, we execute two analyses with the k-fold cross-validation technique by exogenously assuming 10 folds. On the one hand, we find covariates with explanatory power on the different types of dependent variables considering the pair $(\lambda^*_{LOPT}, \alpha^*_{LOPT})$ that minimizes the mean square prediction error (MSPE). On the other hand, we find covariates with explanatory power on the different types of dependent variables considering the pair $(\lambda^*_{LSE}, \alpha^*_{LSE})$ that corresponds to the largest $\lambda$ for the optimal $\alpha$ for which the MSPE is within one standard error of the minimal MSPE. Overall, we substantiate the following conclusions. First, one finds that the former (later) pair contemplates a higher (lower) number of statistically significant covariates, respectively. In particular: total and subscription revenues are explained by 13 covariates when considering the pair $(\lambda^*_{LOPT}, \alpha^*_{LOPT})$, but both are only explained by 5 covariates when considering the pair $(\lambda^*_{LSE}, \alpha^*_{LSE})$, and non-subscription revenue is explained by 10 covariates when considering the pair $(\lambda^*_{LOPT}, \alpha^*_{LOPT})$ and explained by 6 covariates when considering the pair $(\lambda^*_{LSE}, \alpha^*_{LSE})$. Second, one observes that $\alpha^* = 1$ is always verified, which means that the LASSO estimation is unambiguously preferred to ELASTIC NET and RIDGE regressions to explain the different types of dependent variables. Third, knowing that this technique only allows estimating one-day step-ahead coefficients, we conclude that the number of subscribers has a positive, significant, and strong effect on the different types of revenue enjoyed by the online gamer, but remaining covariates exhibit a nearly zero effect on the different types of dependent variables. This implies that both techniques exhibit consistent results that do not necessarily contradict the idea that the revenue of the online gamer increases in redundant magnitude for a stronger intensity of the indirect network effect promoted on viewers, thus, not allowing us to reject the null hypothesis proposed in the identification strategy.

Continual Learning

Results of the linear OLS specification demonstrate that the revenue enjoyed by the online gamer is inversely related to viewership due to the negative sign of the respective coefficient, which is statistically significant at the 1% level. Results of the quadratic OLS specification reveal that all the different types of revenue exhibit an inverted U-shape relationship with viewership, which means that these present similar properties to functions such as the Laffer curve. Therefore, we conclude that there is a critical threshold above which the growth of the network harms any type of revenue enjoyed by the online gamer. As a robustness check, we adopt the Autoregressive Integrated Moving Average (ARIMA) model since this internalizes the optimal time dimension for a given dependent variable. In terms of autoregressive components, results indicate that one-period past values have a positive and significant effect on the first difference of all the possible dependent variables. From a technical point of view, this result is aligned with the idea that CL performs favorably in multi-period forecasting exercises compared to alternative modeling options.

From a substantive point of view, this result suggests that the revenue enjoyed by the online gamer in past periods is expected to have a positive influence over the one enjoyed in the current period.

## 6. Cross-Group Network Externalities and Social Welfare

### 6.1. Impact

The impact of cross-group network externalities on social welfare is analytically untreatable. However, the graphical exposition of simulations provided in Figure 2 allows us to confirm interesting results. Observing panel A, which corresponds to the case where parameters $\sigma_u$, $\sigma_d$, $\gamma$ and $\rho$ are too low:

— Social welfare unambiguously decreases for increasing $\theta$ and $g$ as long as both parameters are sufficiently strong. In other words, if the degree of attraction of members from a given side by members of the opposite side is already considerably high, then a higher intensity in both indirect network effects harms social welfare.
— Social welfare can decrease (increase) for increasing $\theta$ ($g$) as long as both parameters are neither excessively strong nor too weak.
— The asymmetric case characterized by the rise (reduction) of social welfare for increasing $\theta$ ($g$) never holds in equilibrium.
— Social welfare enhances with increasing $\theta$ and $g$ as long as both parameters are sufficiently weak. In other words, if the degree of attraction of members from a given side by members of the opposite side is not excessively high, then a higher intensity in both indirect network effects improves social welfare.

Moreover, panel B reveals the impact of both cross-group network externalities on the equilibrium of social welfare when parameters $\sigma_u$, $\sigma_d$, $\gamma$, and $\rho$ stand in a low-mid range. One concludes that results are similar to those verified in panel A. The unique difference is that one additionally finds the persistence of asymmetric effects on social welfare—in the form of a positive (negative) impact fostered by $g$ ($\theta$), respectively—when $g$ is too low, whatever the value is taken by $\theta$. As such, the presence of a weak capacity of viewers on attracting online gamers boosts asymmetric effects on social welfare according to which a higher $\theta$ ($g$) impacts negatively (positively) on social welfare, respectively.

Indeed, panel C clarifies that the parameter space previously identified becomes larger when parameters $\sigma_u$, $\sigma_d$, $\gamma$, and $\rho$ turn out to have a mid-high range. In addition, panel C indicates that there is no longer room for a reduction of social welfare for increasing $\theta$ and $g$ when both parameters have a sufficiently strong value since the red area disappears. This suggests that the increasing preponderance of demand uncertainty on both sides, market inertia (i.e., risk aversion faced by viewers), and other challenging characteristics (e.g., agreement on the definition of the royalty rate) make a positive impact on both cross-group network externalities more likely to hold in equilibrium. Hence, we claim that the rise of adverse effects is counterbalanced by the positive effect of both cross-group network externalities on social welfare.

Figure 2 also demonstrates that the parameter space where there is the persistence of asymmetric effects on social welfare (which is represented by the blue color) turns out to dominate the parameter space where there is the persistence of unambiguous positive effects on social welfare (which is represented by the green color) once moving from panel C to panel D. This paradigm shift in terms of dominance means that social welfare gains for increasing cross-group network externalities in both sides of the market are less likely to hold when parameters $\sigma_u$, $\sigma_d$, $\gamma$ and $\rho$ are excessively high (i.e., when adverse effects are too strong).

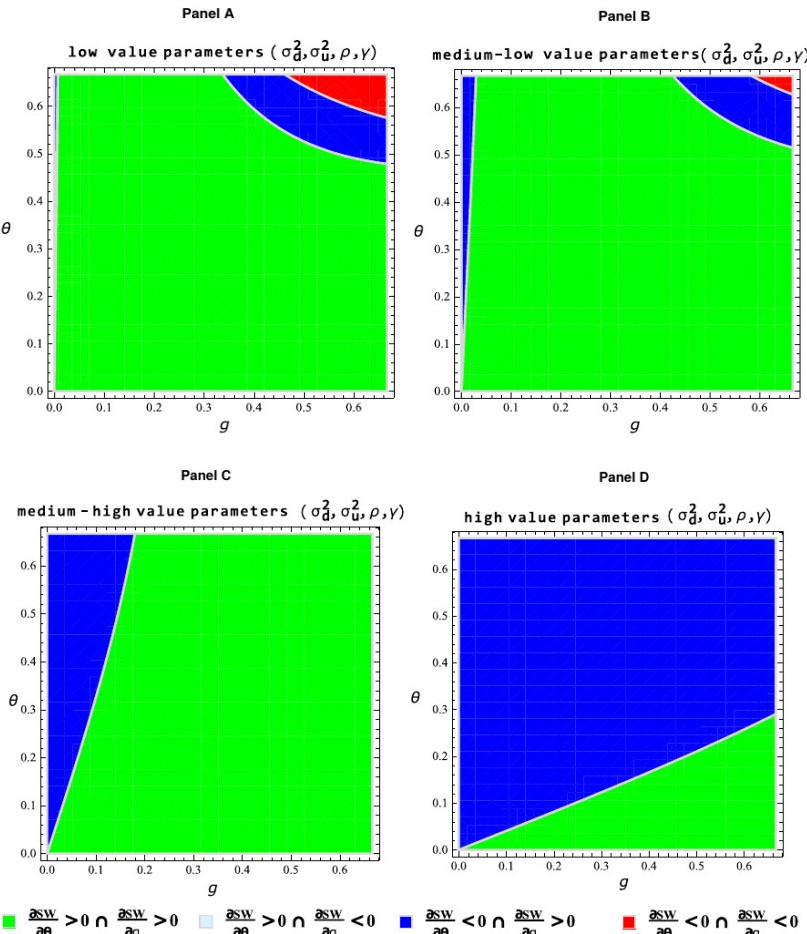

**Figure 2.** Impact of cross-group network externalities on the equilibrium level of social welfare for different values taken by parameters $\sigma_d^2$, $\sigma_u^2$, $\gamma$, and $\rho$. Panel A captures the impact of both cross-group network externalities $\theta$ and $g$ on social welfare when parameters $\sigma_d^2$, $\sigma_u^2$, $\gamma$, and $\rho$ have low value. Panel A simulates the impact of both cross-group network externalities $\theta$ and $g$ when parameters $\sigma_d^2$, $\sigma_u^2$, $\gamma$, and $\rho$ have a low value. Panel B simulates the impact of both cross-group network externalities $\theta$ and $g$ on social welfare when parameters $\sigma_d^2$, $\sigma_u^2$, $\gamma$, and $\rho$ have a low-medium value. Panel C simulates the impact of both cross-group network externalities $\theta$ and $g$ on social welfare when parameters $\sigma_d^2$, $\sigma_u^2$, $\gamma$, and $\rho$ have a medium-high value. Panel D simulates the impact of both cross-group network externalities $\theta$ and $g$ on social welfare when parameters $\sigma_d^2$, $\sigma_u^2$, $\gamma$, and $\rho$ have a high value.

### 6.2. Discussion

One should briefly reflect on the role of both cross-group network externalities on social welfare in light of the academic debate concerning whether the growth and professionalization of online gaming activities are beneficial for society. All over the world, millions of individuals, in particular young men, are currently engaged in online gaming activities, which is far more enticing than in any other historical period. The coordination between online gamers is impressive since these hold the ability to form a strong teamwork to achieve common targets, but the problem is that online gaming is a clear strategic substitute for real jobs. The level of addiction can be so high that a considerable proportion of young men are moving away from the labor market. Figure 2 indicates that the mutual attraction between viewers and online gamers can have either positive or negative effects on social welfare.

On the one hand, this study details conditions whereby the growth of the network related to the online gaming industry is socially desirable. Social welfare gains are expected to prevail when the uncertainty of demand formation in both sides of the market

is high, when a well-established audience of viewers is extremely averse to the online gaming experience and when the royalty rate imposed by the platform to online gamers is sufficiently high as long as online gamers hold the ability to attract additional members from the opposite side of the market. This result suggests that genuine life satisfaction closely linked to the feeling of productivity gains in a similar vein to that experienced in traditional jobs and personal relationships are likely to persist for a limited time period in the context of online gaming. In particular, conditions, whereby social welfare is positively related to both cross-group network externalities, are expected to prevail mostly in the early rollout of engagement of viewers and online gamers in this industry. As such, costs associated with the network growth of the online gaming industry are expected to surpass respective benefits in the long-run perspective.

On the other hand, this research provides a contribution to the salesforce compensation literature by detailing conditions for the persistence of detrimental effects on social welfare caused by the network growth associated with the online gaming industry. At the society level, undesirable effects prevail when the demand uncertainty in both sides of the market is negligible, when a well-established audience of viewers is not excessively averse to the online gaming experience and when the royalty rate imposed by the platform to online gamers is not too high. These conditions combined with strong indirect network effects promoted by both sides of the market on agents from the opposite group imply that social welfare is permanently harmed. Under this circumstance, regulatory policies should be focused on measures aimed at modifying the market environment (e.g., promote the risk aversion of viewers by increasing incentives of engagement in alternative leisure activities, the imposition of ceilings in the number of playing hours, intervention on the royalty rate applied to online gamers). These conclusions are consistent with the need for public governance in private platforms as observed in [23] and the promotion of measures of actuation similar to those described in [10].

Finally, the permanent social welfare loss for increasing cross-group network externalities can also be contextualized in light of the recent event affecting YouTube during the current period of SARS-CoV-2 propagation. This platform unilaterally decided to remove all the online content explicitly disseminating theories that COVID-19 was rapidly spread all over the world due to the negative impact of 5G radiation on human health. Bearing in mind the unilateral action taken by YouTube, this research reveals market conditions where the censorship of online content constitutes welfare-enhancing action. As demonstrated by the presence of a red area in two panels of Figure 2, social welfare is harmed when the network built upon the monopoly-holding platform increases in both sides of the market based on either non-credible news or unreliable theories absent of scientific validation since these are extremely prone to be disseminated when parameters $\sigma_u$, $\sigma_d$, $\gamma$, and $\rho$ are sufficiently low. This is equivalent to saying that the censorship of online content promoted by YouTube can be legitimized as a means of preventing social welfare loss when the demand uncertainty faced by the platform on both sides of the market is negligible. This is true because YouTube already has a maturity of 15 years, when the risk aversion of viewers to absorb new content is redundant, which is true due to the persistence of strong obfuscation on the median viewer, and when the royalty rate applied to content providers is not excessively high. Mutatis mutandis, similar argumentation can be applied to Facebook's action of eliminating comments of President Trump about the impact of COVID-19 on children as well as the unilateral decision taken by the USA on forcing the sale of TikTok's American business to a domestic buyer.

## 7. Conclusions

This study provides a contribution to the salesforce compensation literature by analyzing the effect of cross-group network externalities on equilibrium outcomes in a two-sided market structure where a monopoly-holding platform ensures the interconnection between viewers and online gamers and hires a manager to dissuade the bilateral uncertainty on

membership. The private equilibrium substantiates several conclusions, which can be summarized as follows.

Firstly, one demonstrates the theoretical result that, contrary to the view sustained by canonical studies in two-sided markets, the surplus enjoyed by online gamers and the profit obtained by the monopoly-holding platform may not be strictly increasing for a stronger intensity of the indirect network effect promoted by online gamers on viewers. Contrary to equilibrium outcomes holding in competitive markets, an increasing gap between indirect network effects in disfavor of online gamers increases the managerial compensation provided by the platform and increases the effort spent by the manager on attracting viewers (i.e., the most valuable side of the market based on his/her perspective), but if and only if the risk aversion of viewers and the demand uncertainty on online gamers are both sufficiently high.

Secondly, knowing that equilibrium results are caused by demand uncertainty of the platform on both sides of the market, an empirical analysis is performed to confirm the persistence of a negative or negligible positive impact of the gamer-on-viewer cross-group network externality on the revenue sustained by a Twitch influencer online gamer, which validates the main theoretical outcome of this study.

Thirdly, one analyzes the impact of both cross-group network externalities on social welfare to reveal that society may or may not be better off with the growth and professionalization of online gaming activities. Overall, social welfare effects depend on the value taken by representative parameters of platform's uncertainty on the number of agents that get on board, risk aversion of viewers, and a royalty rate of the monopoly-holding platform applied to online gamers.

Last but not least, this study provides a discussion and several recommendations to ensure the persistence of best regulatory practices in the new and disruptive IoT era. Despite the effort to improve the state-of-the-art of the online gaming and salesforce compensation literature in the context of two-sided markets, this research is not exempted from limitations. Future theoretical avenues include understanding whether similar findings prevail for compensation plans characterized by indirect metrics. The empirical analysis relies on a multivariate time series dataset, which implies that panel data analysis is a natural extension. Artificial neural network architectures can also be implemented to improve the accuracy of estimations. Moreover, the neoclassical microeconomic analysis presented in this study, which relies on a static game theory approach, can be further extended to accommodate more modern approaches such as dynamic market analysis.

**Supplementary Materials:** Mathematical details and additional content on the empirical analysis are available online at https://www.mdpi.com/0718-1876/16/4/40/s1. Supplementary Material is not exposed in the main text only for the sake of brevity. V.M.R. provides data and codes related to this study in his personal GitLab's webpage.

**Author Contributions:** Conceptualization, V.M.R.; Methodology, V.M.R.; Software, V.M.R.; Validation, V.M.R.; Formal Analysis, both authors; Investigation, both authors; Resources, V.M.R.; Data Curation, V.M.R.; Writing—Original Draft Preparation, L.B.; Writing—Review & Editing, V.M.R.; Visualization, V.M.R.; Supervision, V.M.R.; Project Administration, both authors; Funding Acquisition, V.M.R. All authors have read and agreed to the published version of the manuscript.

**Funding:** V.M.R. acknowledges financial support from the project DigEcoBus NORTE-01-0145-FEDER-028540.

**Institutional Review Board Statement:** Not applicable.

**Informed Consent Statement:** Not applicable.

**Data Availability Statement:** Not applicable.

**Acknowledgments:** The authors appreciate valuable suggestions from two anonymous reviewers and Editor-in-Chief, which significantly improved the quality of the study.

**Conflicts of Interest:** The authors declare no conflict of interest.

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
