# Peer review of "Professionalization of Online Gaming? Theoretical and Empirical Analysis for a Monopoly-Holding Platform"

_jtaer, doi:10.3390/jtaer16040040_

Round 1
Reviewer 1 Report
Dear Authors
This is an interesting paper. The strenghts of the paper is that it deals with a highly relevant topic. It is also a hard research challenge to analyse digital markets, and the authors have been successful in making a good analytical model which is tested against empirical data.
The weakness of the paper is that the authors should not only use neo-classical micro-economic analysis (e.g., game theory) for their approach, but also consider more modern approaches, such as dynamic market analysis.
Author Response
See attached file.
Reply to reviewer 1
Dear Authors, this is an interesting paper. The strength of the paper is that it deals with a highly relevant topic. It is also a hard research challenge to analyze digital markets, and the authors have been successful in making a good analytical model which is tested against empirical data. The weakness of the paper is that the authors should not only use neo-classical micro-economic analysis (e.g., game theory) for their approach, but also consider more modern approaches, such as dynamic market analysis.
R: We appreciate the positive feedback given by the reviewer and we totally agree with the report. As such, we include a sentence in the conclusions section (lines 1062 – 1064), where we point out the weakness of the paper and the possibility to further extend the theoretical paper presented by introducing a dynamic setting in future research.
Without further comments, we can only appreciate your suggestions and comments to improve the quality of the manuscript.
Yours sincerely,
The authors.

Reviewer 2 Report
I read carefully the paper entitled: ”Professionalization of online gaming? Theoretical and empirical analysis for a monopoly-holdingplatform”. Although the paper has a certain scientific level, we notice some weaknesses:
- in the introductory part, the connection between the topic of the paper and the purpose and objectives of the journal should be better highlighted / J. Theor. Appl. Electron. Commer. Res.
- bibliographic references to be made according to the model:
”The commercialization of the Internet and the rapid advancement of information technology (IT) have rapidly changed the business landscape [1]. Mobile applications (mobile Apps) have become- increasingly popular in developed countries and in most developing markets due to the extraordinary
growth and development of the smartphone market. Consequently, a new economic landscape, entitled the App economy, has emerged [2].”
References
- Mehra, A.; Paul, J.; Kaurav, S. Determinants of mobile apps adoption among young adults: Theoretical extension and analysis. J. Mark. Commun. 2020, 1–29.
- Kim, W.; Kankanhalli, A.; Lee, L. Investigating decision factors in mobile application purchase: A mixed-methods approach. Inf. Manag. 2016, 53, 727–739.
- the structuring and technical editing of the paper does not follow the rules of J. Theor. Appl. Electron. Commer. Res.
- Idem. Graph notations.
- the authors exaggerate in the use of footnotes;
- to rethink the volume and structure of the paper (the authors exaggerate by extending the explanations and demonstrations / calculations / mathematical relationships, because there is a risk of weakening readers' interest in this content);
If they read carefully the text of the paper, the authors will detect for themselves the negligences ...
But he must apply the rules of the Journal.
It would be good to expand the bibliography with some articles from prestigious scientific journals (WoS) published in 2019 and 2020. All in all, eventually, it may be seen by an English teacher (native).
Author Response
Reply to reviewer 2
- In the introductory part, the connection between the topic of the paper and the purpose and objectives of the journal should be better highlighted
The comment of the reviewer is appropriate. To correct this failure, we introduce lines 130-139, where we say that this study aims at filling a research gap related to the share and debate of new ideas in an emerging and rapidly evolving two-sided market. Business practices, social, cultural, and legal concerns, personal privacy and security, communications technologies, and social welfare effects resulting from the professionalization of online gaming are among the most relevant elements of a new digital age characterised by distinct employment opportunities compared to the period prior to the Internet of Things (IoT), which may or may not act in the benefit of society.
- Bibliographic references to be made according to the model:
”The commercialization of the Internet and the rapid advancement of information technology (IT) have rapidly changed the business landscape [1]. Mobile applications (mobile Apps) have become- increasingly popular in developed countries and in most developing markets due to the extraordinary growth and development of the smartphone market. Consequently, a new economic landscape, entitled the App economy, has emerged [2].” References
- Mehra, A.; Paul, J.; Kaurav, S. Determinants of mobile apps adoption among young adults: Theoretical extension and analysis. J. Mark. Commun. 2020, 1–29.
- Kim, W.; Kankanhalli, A.; Lee, L. Investigating decision factors in mobile application purchase: A mixed-methods approach. Inf. Manag. 2016, 53, 727–739.
This sentence is now included in the introduction in lines 90-102.
- The structuring and technical editing of the paper does not follow the rules of J. Theor. Appl. Electron. Commer. Res.
- Idem. Graph notations.
The reviewer can now observe that the manuscript satisfies the formal rules of J. Theor. Appl. Electron. Commer. Res.
- The authors exaggerate in the use of footnotes;
The comment of the reviewer is appropriate. The number of footnotes was reduced to 5.
- To rethink the volume and structure of the paper (the authors exaggerate by extending the explanations and demonstrations/calculations/mathematical relationships, because there is a risk of weakening readers' interest in this content); If they carefully read the text of the paper, the authors will detect for themselves the negligence ... But he must apply the rules of the Journal. It would be good to expand the bibliography with some articles from prestigious scientific journals (WoS) published in 2019 and 2020. All in all, eventually, it may be seen by an English teacher (native).
The reviewer can now observe that the paper follows the formal rules of J. Theor. Appl. Electron. Commer. Res. In addition, the paper now includes additional bibliography from papers published in 2019-2020 in WoS journals (see lines 131-133) and the language employed in the text of the entire paper has been revised by a native English professional. We also reduce the size of the paper.
Without further comments, we thank you in advance for the suggestions and help on improving the quality of our study. Yours sincerely,
The authors.

Round 2
Reviewer 2 Report
Manuscript has been significantly improved and now warrants publication in JTAER.